# The Synthesis of Bio-Based Michael Donors from Tall Oil Fatty Acids for Polymer Development

**DOI:** 10.3390/polym14194107

**Published:** 2022-09-30

**Authors:** Ralfs Pomilovskis, Inese Mierina, Hynek Beneš, Olga Trhlíková, Arnis Abolins, Anda Fridrihsone, Mikelis Kirpluks

**Affiliations:** 1Polymer Laboratory, Latvian State Institute of Wood Chemistry, Dzerbenes St. 27, LV-1006 Riga, Latvia; 2Institute of Technology of Organic Chemistry, Faculty of Materials Science and Applied Chemistry, Riga Technical University, P. Valdena St. 3/7, LV-1048 Riga, Latvia; 3Institute of Macromolecular Chemistry, CAS, Heyrovského nám. 2, 162 06 Prague 6, Czech Republic

**Keywords:** Michael addition components, Michael donor, tall oil fatty acids, bio-based acetoacetate

## Abstract

In this study, the synthesis of a Michael donor compound from cellulose production by-products—tall oil fatty acids—was developed. The developed Michael donor compounds can be further used to obtain polymeric materials after nucleophilic polymerization through the Michael reaction. It can be a promising alternative method for conventional polyurethane materials, and the Michael addition polymerization reaction takes place under milder conditions than non-isocyanate polyurethane production technology, which requires high pressure, high temperature and a long reaction time. Different polyols, the precursors for Michael donor components, were synthesized from epoxidized tall oil fatty acids by an oxirane ring-opening and esterification reaction with different alcohols (trimethylolpropane and 1,4-butanediol). The addition of functional groups necessary for the Michael reaction was carried out by a transesterification reaction of polyol hydroxyl groups with *tert*-butyl acetoacetate ester. The following properties of the developed polyols and their acetoacetates were analyzed: hydroxyl value, acid value, moisture content and viscosity. The chemical structure was analyzed using Fourier transform infrared spectroscopy, gel permeation chromatography, size-exclusion chromatography and nuclear magnetic resonance. Matrix-assisted laser desorption/ionization analysis was used for structure identification for this type of acetoacetate for the first time.

## 1. Introduction

Bio-based polymer development is essential for the European Green Deal to transit into a circular bioeconomy as Europe strives to be climate neutral by 2050 [1]. It is quite clear that incentives will be given to the bio-based polymer industry to encourage it to expand, develop new technologies and materials and increase production quantities [2,3]. The bio-based polymer industry must overcome several challenges, such as changing the raw material base, implementing greener technology, modifying infrastructure and establishing new supply and value chains [4]. It is dire to find solutions to these challenges and to reorganize the carbon processing sector.

Tall oil is obtained as a side stream in pulp production. Tall oil is a second generation feedstock; thus, it does not compete with the food and feed supply chain. It is a complex mixture of tall oil fatty acids, mainly oleic and linoleic acid, rosin acids and unsaponifiables such as high-molecular alcohols, sterols and other alkyl hydrocarbon derivatives. Currently, its potential to produce high value-added bio-based substances and polymeric materials has been investigated relatively little [5]. Fatty acids of tall oil contain a double bond, making them attractive for introducing a reactive functional group and suitable for polymer production [6,7].

The modification of double bonds has been extensively studied. There are many publications on the double bond epoxidation of fatty acids by different methods for further synthesizing polyols by ring-opening reactions with alcohols [8,9,10,11]. In a couple of studies, these polyols are used as a part of component A to produce polyurethanes [3,7,12,13,14,15]. Isocyanates are used as component B in conventional polyurethane production. Isocyanates are considered to be highly toxic and environmentally hazardous substances [16,17,18,19,20]. Although non-isocyanate polyurethanes (NIPUs) are offered as an alternative to conventional polyurethanes and are considered a safe and sustainable way to extract polymers, the NIPU production technology also faces significant difficulties. For NIPUs, production occurs under harsh conditions—high pressure, high temperature and a long reaction time [21]. The industry needs to find more ways to obtain polymers under mild conditions. It is possible to obtain a two-component thermoset polymeric material with similar properties to polyurethane materials by exploiting the nucleophilic 1,4-addition reaction, which does not involve the application of isocyanates. Hydroxyl groups of polyols can be converted to β-ketoesters to obtain polymers with acrylates by the Michael reaction. Although the method is known, the research has mainly been conducted for coating materials [12,22,23,24,25].

Bio-based polymer development using β-ketoesters is an emerging field. Therefore, the acetoacetylation of natural fatty acids and their polyols is not broadly studied yet. A limited number of studies on acetoacetylation of bio-based polyols have been published. Acetoacetylation has been successfully demonstrated in the synthesis of soybean-oil-based polyols [12], castor oil and castor-oil-based polyols [22,23,24,25,26,27]. Hydroxyl groups of fatty acids polyols are relatively easy to acetoacetylate with *tert*-butyl acetoacetate by a transesterification reaction, thereby obtaining β-ketoesters. The reaction is mostly carried out at a temperature of 110–130 °C [12,22,23,24,28,29].

It is currently believed that *tert*-butyl acetoacetate is the most suitable reagent for the transacetoacetylation reaction due to its effectiveness under relatively mild conditions, in contrast to other analogs such as methyl acetoacetate, ethyl acetoacetate or isopropyl acetoacetate. Additionally, *tert*-butyl acetoacetate has several advantages, such as stability in storage, a relatively low cost and wide commercial availability, making it attractive from an industrial point of view [29].

In addition, the viscosity of polyol significantly decreases after acetoacetylation due to the disappearance of intermolecular hydrogen bonds. Thus, the obtained β-ketoesters are more easily handled in industrial products [12,22,27,30,31]. In the case of tall oil, lower viscosity is a particularly notable benefit for industrial use, as tall oil fatty acids polyols have a very high viscosity. It can be even above 100,000 mPa·s [7].

This study explores the suitability of cellulose production side stream—tall oils—to provide new insight into bioresource-based feedstocks for further bio-based polymer production. A Michael donor was synthesized, one of the components for a two-component system that cures at room temperature after mixing, forming a highly cross-linked and mechanically rigid polymer. It is a completely new way of valorizing a second generation feedstock—a tall oil.

For the first time, different tall-oil-based polyols were used to synthesize Michael donors by converting the hydroxyl groups present in polyols into acetoacetate esters. Polyols were synthesized from tall oil by catalytical epoxidation using ion exchange resin and subsequent cleavage of the oxirane ring and esterification of the acid group with polyfunctional alcohol. Acetoacetates were synthesized from two tall oil polyols and two commercially available polyols. The following properties of the developed polyols and their acetoacetates were analyzed: hydroxyl value, acid value, moisture content and viscosity. The chemical structure was determined using gel permeation chromatography and size-exclusion chromatography (GPC/SEC), Fourier-transform infrared spectroscopy (FTIR), matrix-assisted laser desorption/ionization (MALDI-TOF) spectra and nuclear magnetic resonance (NMR).

## 2. Materials and Methods

### 2.1. Materials and Reagents for Synthesis and Analysis

Reagents for synthesis: tall oil fatty acids (TOFA) (trade name “FOR2”), the content of TOFA: fatty acids > 96%, rosin acids 1.9% and unsaponifiables 1.8% from Forchem Oyj, Rauma, Finland; Amberlite IR-120 H, strongly acidic, hydrogen form from Fluka; glacial acetic acid, ACS reagent, ≥99.7%; hydrogen peroxide, purum p.a., ≥35%; *tert*-butyl acetoacetate, reagent grade, 98%; trimethylolpropane (TMP), reagent grade, 97%; and 1,4-butanediol (BD), reagent plus, ≥99% were purchased from Sigma-Aldrich, Schnelldorf, Germany; tetrafluoroboric acid solution, 48 wt.% in water, was purchased from Alfa Aesar, Kandel, Germany. Lupranol 3300 (L3300), hydroxyl value is 400 mg KOH/g, trifunctional, was purchased from BASF, Lemförde, Germany. Neopolyol 380 (NEO380), has a hydroxyl value of 370 mg KOH/g and a functionality of 3.3, purchased from NEO Group, Rimkai, Lithuania.

Reagents for analysis: 4-(dimethylamino)pyridine, reagent plus, ≥99%; acetic anhydride, puriss, ≥99%; anhydrous sodium sulfate, ACS reagent, granular, ≥99.0%; chloroform, puriss p.a., 99.0–99.4%; dichloromethane, puriss p.a., ACS reagent, ≥99.9%; methanol, puriss p.a., ACS reagent, ≥99.8%; N,N-dimethylformamide, ACS reagent, ≥99.8%, water content ≤150 ppm; perchloric acid, ACS reagent, 70%; potassium hydroxide, ACS reagent, pellets, ≥85%; potassium iodide, ACS reagent, ≥99%; tetraethylammonium bromide, reagent grade, 98%; 2,5-dihydroxybenzoic acid, ≥98%; sodium trifluoroacetate ≥99.5%; N,N-dimethylformamide, biotech. grade, ≥99.9%, purchased from Sigma Aldrich, Schnelldorf, Germany. Hanus solution, volumetric 0.1 M IBr, and sodium thiosulphate fixanals 0.1 M were purchased from Fluka, Seelze, Germany. Tetrahydrofuran, puriss p.a., 99.9%, was purchased from Lachner, Neratovice, Czech Republic.

### 2.2. Synthesis of Acetoacetylated Tall Oil Fatty Acid Polyols

#### 2.2.1. Epoxidation of TOFA

The mixture of TOFA (700 g), acetic acid (126 g) and ion exchange resin (IR) Amberlite IR-120 H (20 wt.% of TOFA) was heated to 40 °C in a four-neck round flask, which was immersed in a thermostatic water bath. An inert environment was provided by argon gas. Then, slowly and evenly, the hydrogen peroxide (614 g) was added through the dropping funnel while watching the temperature; therefore, it did not exceed 60 °C. After adding hydrogen peroxide, the temperature was kept at 60 °C for 6 h, and the medium was stirred at 600 rpm. The obtained mixture was washed with ethyl acetate and warm distilled water (60 °C) and was dried using a rotatory vacuum evaporator to separate the epoxidized tall oil fatty acids (E^IR^TOFA).

#### 2.2.2. Synthesis of Polyols from E^IR^TOFA

Polyols were obtained from the synthesized E^IR^TOFA by opening the oxirane ring and subsequent esterification with alcohols. Two alcohols of different functionality were used for this step: BD and TMP.

E^IR^TOFA (300 g) was added via a dropping funnel to a four-neck flask (1 L) already containing the mixture of the alcohol (amount of E^IR^TOFA carboxyl and oxirane group to alcohol moles ratio of 1:1) and tetrafluoroboric acid solution (48 wt.% in H2O, 0.3 wt.% from E^IR^TOFA) as a catalyst. The flask was immersed in a thermostatic oil bath. The temperature was maintained at 80 °C during the addition of the E^IR^TOFA. In this step, the oxirane ring was opened. After all the epoxidized oil had been added, the temperature was raised to 180 °C. The stirring rate was 500 rpm. The water formed during the synthesis was removed with an inert carrier gas (nitrogen) and was condensed in a Liebig condenser. The duration of esterification was 6 h.

#### 2.2.3. Acetoacetylation of Polyols

The polyol acetoacetylation was performed with *tert*-butyl acetoacetate by a transesterification reaction. Via dropping funnel, *tert*-butyl acetoacetate was added to E^IR^TOFA polyol (150 g) in a three-neck round bottom flask (500 mL). The amount of polyol hydroxyl groups to *tert*-butyl acetoacetate moles ratio was 1:1. The flask was immersed in a thermostatic oil bath, and the temperature was kept constant at about 120 °C throughout the synthesis. The *tert*-Butanol released during the reaction was condensed using a Liebig condenser. Water was used as a cooling agent in the Liebig condenser at a temperature of about 25–30 °C because the melting point of *tert*-butanol is 25 °C. If a lower temperature is used for cooling, there is a possibility that *tert*-butanol crystallizes in the cooler. The intense *tert*-butanol release ended after about 2 h. In total, the reaction was performed for 4 h to maximize conversions.

The amount of acetoacetate groups (*AA*) in the synthesized Michael donors was calculated according to Equation (1):(1)$$AA_{groups}=\frac{n_{tert-\mathrm{butyl}\;\mathrm{acetoacetate}}}{m_{yield}}\cdot 100$$
where *n_tert_*_-butyl acetoacetate_ is used for *tert*-butyl acetoacetate (mol), and *m_yield_* is the product mass (g).

As a result of this step, the following Michael donors were obtained:epoxidized tall oil fatty acids 1,4-butanediol polyol acetoacetate (E^IR^TOFA_BD_AA);epoxidized tall oil fatty acids trimethylolpropane polyol acetoacetate (E^IR^TOFA_TMP_AA);Neopolyol 380 acetoacetate (NEO380_AA);Lupranol 3300 acetoacetate (L3300_AA).

### 2.3. Characterization of the Synthesized Michael Donor Components

All titration methods were used in accordance with the testing standards. The acid value and hydroxyl value were determined according to ISO 2114:2000 and ISO 4629-2:2016 standards, respectively. ISO 3961:2013 and ASTM D1652-04:2004 testing standards were used for epoxy and iodine value determination, respectively. The moisture content was measured using Karl Fischer titration using the Denver Instrument Model 275 KF automatic titrator (Denver Instrument, Bohemia, NY, USA).

The rheological measurements were made using the Anton Paar Modular Compact Rheometer MCR 92 (Anton Paar, Graz, Austria) with a cone-plate measuring system and a gap of 48 μm. Shear rate ramps were carried out from 1 to 1000 s^−1^. The measurements were performed at a temperature of 25 °C, which was kept constant by the temperature hood.

The synthesized monomer structure was analyzed by FTIR. The spectra were registered with a Thermo Scientific Nicolet iS50 spectrometer (Thermo Fisher Scientific, Waltham, MA, USA) at a resolution of 4 cm^−1^ (32 scans). The FTIR data were collected using the attenuated total reflectance technique with diamond crystals. A drop of the synthesized components was dripped directly onto the prism and analyzed.

The NMR spectra for the samples were recorded on a Bruker spectrometer (Bruker BioSpin AG, Fällanden, Switzerland) at 500 MHz and 126 MHz for ^1^H and ^13^C spectra, respectively.

MALDI-TOF mass spectra were acquired with the UltrafleXtreme TOF–TOF mass spectrometer (Bruker Daltonics, Bremen, Germany) equipped with a 2000 Hz smartbeam-II laser (355 nm) using the positive ion reflectron mode. Panoramic pulsed ion extraction and external calibration were used for molecular weight assignment.

The dried droplet method was used, where the solutions of the sample (10 mg mL^−1^), matrix 2,5-dihydroxybenzoic (20 mg mL^−1^) and ionizing agent sodium trifluoroacetate (CF_3_COONa; 10 mg mL^−1^) in N,N-dimethylformamide were mixed in the volume ratio of 4:20:1. The mixture (1 mL) was deposited on the ground steel target.

SEC analyses were carried out in tetrahydrofuran at 25 °C and with a flow rate of 1 mL min^−1^ using a GPC system equipped with a refractive index detector (Shodex, Tokyo, Japan). A set of three PLgel columns with a particle size of 10 μm and pore sizes of 104, 103 and 50 Å, 300 × 7.5 mm (Polymer laboratories, Church Stretton, UK) were used. Polystyrene standards were used for calibration.

## 3. Results and Discussion

Ion exchange resin, an easily recyclable and reusable catalyst, was applied in the synthesis process as an alternative to commonly used inorganic acid catalysts [32]. Moreover, no solvent was used in the syntheses, nor were excess intermediates obtained, such as tall oil methyl esters. It made the synthesis process “greener” and more straightforward with fewer steps. A schematic overview for the synthesis of tall-oil-based Michael’s donor components is shown in Figure 1.

### 3.1. Characteristics of Synthesized Michael Donor Components

Synthesized acetoacetates and intermediates were characterized using different titration methods to determine acid, hydroxyl, epoxy and iodine values. The chemical structure was analyzed using GPC/SEC, FTIR and MALDI-TOF spectra. The viscosity and its dependence on the applied shear rate were also determined. Table 1 lists an overview of the characteristics determined using titrimetric methods and can be used to describe the synthesis process.

For E^IR^TOFA, the acid value decreased significantly compared to non-epoxidized tall oil. This was mainly because by-products were formed during the epoxidation process by oxirane ring opening with a carboxyl group of fatty acid to form dimers and even trimers. The GPC/SEC graphs and MALDI-TOF spectra confirmed the formation of by-products. The amount of double bonds significantly reduced during epoxidation, which indicated an effective conversion of double bonds into oxirane. The E^IR^TOFA oxirane value was 2.142 mmol/g. The obtained polyols had a negligible acid value compared to epoxidized oil, which indicated a good yield of the esterification reaction.

Two different polyfunctional alcohols were used to introduce hydroxyl groups into the tall oil fatty acid structure and to design E^IR^TOFA polyols with varied hydroxyl group functionality. Due to different oxirane-opening reagents, the hydroxyl value of the obtained E^IR^TOFA polyols differed almost twice. The higher hydroxyl value was for a polyol obtained with a trifunctional TMP oxirane-opening reagent. The use of BD as an oxirane-opening reagent yielded a smaller hydroxyl value. Synthesized bio-polyols allow one to obtain bio-based polymers with varied physical properties further.

After acetoacetylation, the hydroxyl number decreased significantly as the hydroxyl groups were replaced with the β-ketoester groups for E^IR^TOFA-based polyols. The same relationship was observed for commercial polyols, which were selected to compare the synthesis process, the properties of the products and the suitability for the further extraction of polymeric materials. The content of acetoacetate groups was determined by calculations from the amounts of used acetoacetate and polyol.

### 3.2. Viscosity Analysis of the Michael Donor Components

Viscosity was measured for TOFA, E^IR^TOFA and E^IR^TOFA-based polyols and subsequently synthesized acetoacetates and for commercial polyols and their acetoacetates. The results of the rheological measurements are presented in Figure 2.

The viscosity did not depend on the applied shear rate, indicating that the obtained substances were Newtonian fluids. The Newtonian behavior of the developed Michael donors greatly facilitates the use of these acetoacetates in the industry, as the effect of the applied shear rate on the viscosity properties does not have to be taken into account.

The measured viscosities are summarized in Table 2. As can be seen, the viscosity of E^IR^TOFA was significantly higher than that of TOFA. Firstly, the increased viscosity was caused by the formation of by-products (dimers and oligomers) with increased average molecular weight. Secondly, the opening of the oxirane ring resulted in the formation of hydroxyl groups that led to an intermolecular hydrogen bonding network.

The most significant increase in viscosity was observed for the E^IR^TOFA_TMP polyol that was synthesized by opening the oxirane ring and by the esterification of E^IR^TOFA with TMP. The viscosity of E^IR^TOFA_TMP was higher than 118,000 mPa·s, which is considered to be relatively high viscosity and makes it difficult to use the product in the industry. E^IR^TOFA_TMP polyol exhibits such a high viscosity mainly due to increased intermolecular hydrogen bonding. However, after E^IR^TOFA_TMP polyol was acetoacetylated, the viscosity significantly decreased to 5,500 mPa·s. Therefore, the E^IR^TOFA_TMP_AA is more suitable for industrial use.

Compared to commercial or bio-based polyols, acetoacetates showed a significant decrease in viscosity due to the decrease in hydroxyl groups and hence the disappearance of intermolecular hydrogen bonds [12,22,27,30,31]. This significantly improves the potential use of bio-polyol acetoacetates in the industry. In the adjustment of existing equipment, in pump capacity alteration and in easier transportation through pipelines, they exhibit better flowability compared to neat polyols from the same raw material.

### 3.3. FTIR Analysis of the Michael Donor Components

FTIR spectra of TOFA, synthesized polyols, commercial polyols and acetoacetylated polyols are shown in Figure 3. The absorption bands of the FTIR spectrum characterize functional groups very well. The course of synthesis and the changes associated with the functional groups of the molecules can be determined.

A new peak appeared at 831 cm^−1^ in the FTIR spectrum after the epoxidation of TOFA, which corresponded to the vibrations of the oxirane ring (Figure 3a). This indicated that the epoxidation in the presence of ion exchange resin Amberlite IR-120 H was successful. Transformations of double bonds were also indicated by the peaks at ~1654 cm^−1^ and ~3009 cm^−1^, corresponding to C=C stretching vibrations and =C-H stretching vibrations, respectively. The intensities of these peaks for E^IR^TOFA decreased. The spectrum of E^IR^TOFA also showed the appearance of a new, relatively weak but noticeable absorbance band between 3600 and 3150 cm^−1^, which typically characterizes vibrations of the hydroxyl group. This indicated that the opening of the oxirane ring occurred as an undesirable side reaction during the epoxidation process. As seen in the spectrum, the intensity of the absorption band of the carbonyl group of the acid moiety at ~1705 cm^−1^ decreased mainly due to ester formation, resulting in undesired by-products. Hydroxyl groups and esters were formed from the cleavage of the oxirane ring in the reaction with the carboxyl group of tall oil or acetic acid.

The absorption of the hydroxyl group band significantly increased for synthesized polyols. Moreover, this band was more intense for the polyols obtained by the E^IR^TOFA reaction with TMP alcohol because the TMP polyol contained more hydroxyl functional groups. The spectrum also reflected the disappearance of the oxirane ring absorbance band at ~831 cm^−1^. A decrease in the intensity of the peaks of the acid group stretching vibration and an increase in the ester carbonyl group at ~1738 cm^−1^ indicated that the esterification reaction was successful.

In the FTIR spectra, the intensity of the hydroxyl group stretching vibration absorbance bands decreased significantly for acetoacetylated polyols. They were nearly imperceptible. This is a very important indicator of a successful acetoacetylation reaction. The spectrum of acetoacetylated polyols showed an increase in the intensity of the carbonyl stretching vibration peak of ester at ~1738 cm^−1^ and ketone at ~1715 cm^−1^. The appearance of a relatively strong, new absorption band between 1670 and 1580 cm^-1^ was observed. Acetoacetates exhibited keto-enol tautomerism, and the carbonyl group appeared at a lower frequency due to intramolecular hydrogen bonding in the enol form. The same changes in the FTIR spectra were also observed for the acetoacetylated commercial polyols (Figure 3b).

### 3.4. MALDI-TOF Spectra of the Michael Donor Components

The MALDI-TOF spectra of synthesized polyols and their acetoacetates are shown in Figure 4. The spectra show that a mixture consisting of various compounds was formed in the epoxidation process and in the polyol synthesis and their subsequent acetoacetylation.

The spectrum was also taken for E^IR^TOFA, but it was difficult to interpret because the peak corresponding to epoxidized oleic and linoleic acids overlaps with the peaks of the matrix. However, the spectrum of E^IR^TOFA clearly shows that dimers and trimers were formed as by-products in the epoxidation process. The dominant monomers of synthesized polyols and their acetoacetates are clearly visible in the spectrum. Dimers in the case of E^IR^TOFA_BD polyol and its corresponding acetoacetate were identified in the MALDI-TOF spectrum (see Figure 4a).

The MALDI-TOF spectra show that the acetoacetylation process was successful. After acetoacetylation, the most intense polyol peaks shifted by such units of m/z that correspond to the increase in mass, which corresponds to the replacement of all hydroxyl groups of the molecule with the acetoacetate group. For example, the characteristic peak of epoxidized oleic acid TMP polyol appeared at 571 m/z [C_30_H_60_O_8_^+^Na]^+^. After acetoacetylation, the peak of this polyol acetoacetate shifted to 991 m/z (Figure 4b). The difference of 420 m/z corresponds exactly to the molar mass of the five acetoacetate groups and fully correlates with synthesized polyol functionality.

As the peaks of dimers, trimers and the other by-products were shifted after acetoacetylation, an acetoacetate group was successfully introduced into the molecules. It can be concluded that these by-products were also fully suitable for producing polymer materials by the Michael reaction. Increased viscosity could adversely affect these dimers, trimers and by-products due to the large and branched structure. For more information on the possible corresponding structures of the intense peaks in the MALDI-TOF spectra (Figure 4), see Appendix A Appendix A.

The MALDI-TOF spectra of NEO380 polyol (Figure 4c) show that the most intense peak appeared at 513 m/z after acetoacetylation. The difference (168 m/z) corresponds to two acetoacetate groups, which indicates that this compound was difunctional and contained two hydroxyl groups.

Relatively few peaks appeared in the MALDI-TOF spectrum of L3300 polyol (Figure 4d) and its acetoacetate, compared to bio-polyol acetoacetate and NEO380_AA spectra. L3300 is a trifunctional polyether polyol based on glycerine, and 2-hydroxypropoxy groups are added to the glycerol molecule by oxypropylation in industrial synthesis.

The L3300 is a relatively pure substance, making the spectrum easier to interpret. The most intensive characteristic peaks of the compounds were with an interval of 58 m/z, which corresponds to the molar mass of the 2-hydroxypropoxy group. This spectrum distinctly shows how evenly the oxypropylation had taken place. As shown in the spectrum in Figure 4d, after acetoacetylation, all peaks shifted by 252 m/z, corresponding to the molar mass of the three acetoacetate groups. This number is compatible with the free hydroxyl groups of the molecule.

### 3.5. GPC/SEC Analysis of the Michael Donor Components

The GPC/SEC chromatograms of the synthesized acetoacetates are shown in Figure 5. According to the chromatogram, the ion exchange resin catalyzed epoxidation process of TOFA produced a significant number of by-products (Figure 5a). The formation of dimers, trimers and other oligomers and oxirane cleavage products are clearly visible in the spectrum. Abolins et al. also identified similar by-product formation in the epoxidation of TOFA [7]. The retention time for the oxirane cleavage products was ~23.5 min, but dimers and trimers were identified at retention times of ~22.6 and ~22.0 min. The chromatogram of polyols showed that the peaks shifted, and their retention times decreased. This indicated an increase in molecular weight.

In the spectrum of E^IR^TOFA_BD polyol, a new peak appeared at a retention time of ~25.4 min, which characterizes the free unreacted BD. In the case of E^IR^TOFA_TMP polyol, the free TMP peak appeared at the retention time of ~23.7 min. After acetoacetylation, the molecular weight increased, thus reducing the retention times. It is distinctly visible in the chromatogram. The results of GPC/SEC correlate with the results of the MALDI-TOF analyses.

Although synthesized E^IR^TOFA-based polyols consist of a mixture of various derivatives, they contain free hydroxyl groups, which can be converted to β-ketoesters. The obtained substances with many acetoacetate groups are suitable for further use in polymer development.

A significant disadvantage of oligomerization by-products is that they are highly branched with a high molecular weight, resulting in a higher average molecular weight and viscosity. The TOFA epoxidation method can be improved to reduce the formation of oligomerization by-products. However, on the other hand, the obtained bio-based acetoacetates are fully suitable for the production of the polymer material by the Michael reaction. Polymers with higher cross-link densities can be obtained from dimers, trimers and oligomers, and it may be worth exploring their effect on cross-link density in detail.

The commercial polyol NEO380 also consists of a mixture of several components. All peaks of NEO380 shifted after acetoacetylation, and the retention time decreased (Figure 5b), indicating that most of the compounds in this mixture contained an acetoacetate group after acetoacetylation.

There is only one pronounced peak in the L3300 polyol spectrum (Figure 5b). There was a clearly identifiable shift in the spectrum of products, indicating an increase in molecular weight after acetoacetylation.

### 3.6. NMR Spectra of the Michael Donor Components

^1^H NMR spectra clearly demonstrate (see Figure 6) the formation of the epoxide moiety (signals at 2.8–3.2 ppm), although partial cleavage of the oxirane moiety occurred (signals at 3.4–4.1 ppm). The above-mentioned results (see Figure 3, Figure 4 and Figure 5) of the simultaneous cleavage of the oxirane ring and acid-catalyzed esterification were smoothly accompanied by NMR spectra, as well. As a result of the oxirane ring cleavage, the signals (^1^H-NMR: 2.84–3.14 ppm) raised from epoxide both disappeared when the BD and TMP were used for cleavage. New signals, characteristic of the -O-CH_2_- group at 4.0–4.5 ppm, were found (see Figure 7). The formation of the ester bonds with BD and TMP was confirmed by ^13^C spectra, as well; the signal of the carbonyl group shifted from 180 ppm (for the carboxylic acid) to 174 ppm (for the ester). The polysubstituted derivatives of TMP in small quantities were also observed. The last one was deducted from several small signals at 65.5–63.5 ppm (-CH_2_- from TMP moiety) and 43–42 ppm (quaternary carbon from TMP moiety) in the ^13^C spectrum.

Several changes in the NMR spectra confirmed the successful acetoacetylation of the bio-based polyols and commercial polyols NEO380 and L3300 (see Figure 8). The signal characteristic of the methylene moiety of the acetoacetyl group appeared at 3.4 ppm. Clear signals assigning both the ketone (at 200 ppm) and the acetoacetic acid ester (at 167 ppm) moieties were detected in the ^13^C spectra.

## 4. Conclusions

This paper reports the synthesis and characterization of different Michael donor components. For the first time, different tall-oil-based polyols were used to synthesize Michael donors. These donors were synthesized from bio-based polyols that were obtained from epoxidized tall oil fatty acids by oxirane ring-opening and esterification with alcohols and subsequent hydroxyl group acetoacetylation with *tert*-butyl acetoacetate by a transesterification reaction.

The successful synthesis of Michael donors from the cellulose production side stream—tall oil fatty acids—and commercial polyols was confirmed by FTIR, GPC/SEC, MALDI-TOF and NMR data. Moreover, after the polyols were acetoacetylated, the viscosity significantly decreased and made it more suitable for further use. Although synthesized substances are a mixture of various components—different monomers, dimers and other oligomers—they can be used as promising reagents for the synthesis of polymeric materials by the Michael reaction polymerization.

Different chemical structures and functionalities of the synthesized bio-based components can allow for the development of polymer formulations with varied cross-link density. In this study, the investigated tall oil and commercial polyol-based Michael donors can allow for the development of polymer formulations suitable for two-component polymer foams, coatings, resins and composite matrices. The obtained Michael donors were used by Pomilovskis et al. [33] for polymer synthesis.

## Figures and Tables

**Figure 1 polymers-14-04107-f001:**
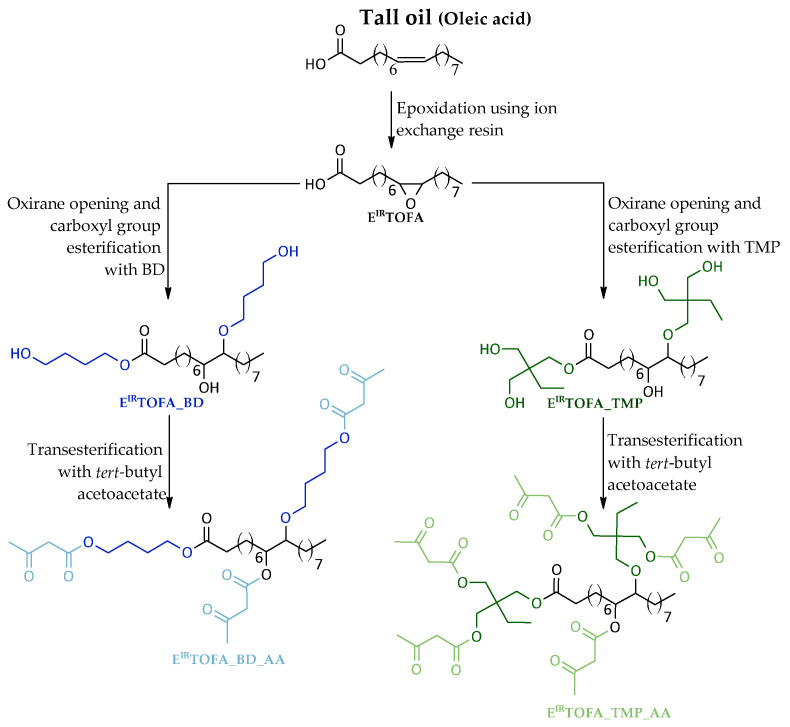
Idealized scheme of TOFA (oleic acid)-based Michael donor development.

**Figure 2 polymers-14-04107-f002:**
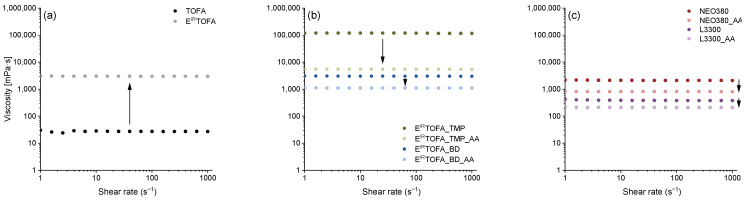
Rheological viscosity versus the shear rate at 25 °C for: (**a**) TOFA and E^IR^TOFA; (**b**) E^IR^TOFA-based polyols and their acetoacetates; (**c**) commercial polyols and their acetoacetates.

**Figure 3 polymers-14-04107-f003:**
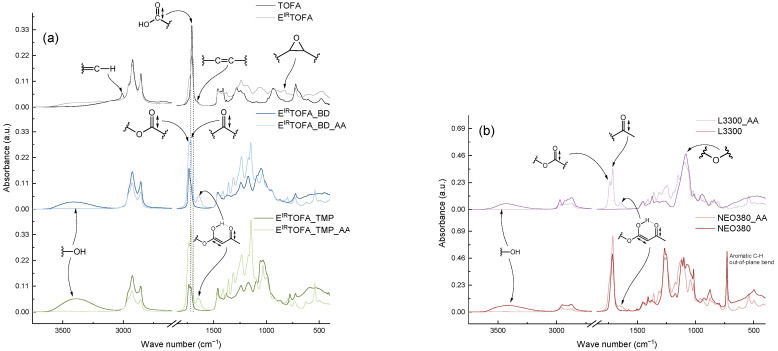
FTIR spectra of TOFA, E^IR^TOFA, synthesized polyols, commercial polyols and acetoacetylated polyols: (**a**) based on tall oils; (**b**) based on commercial polyols.

**Figure 4 polymers-14-04107-f004:**
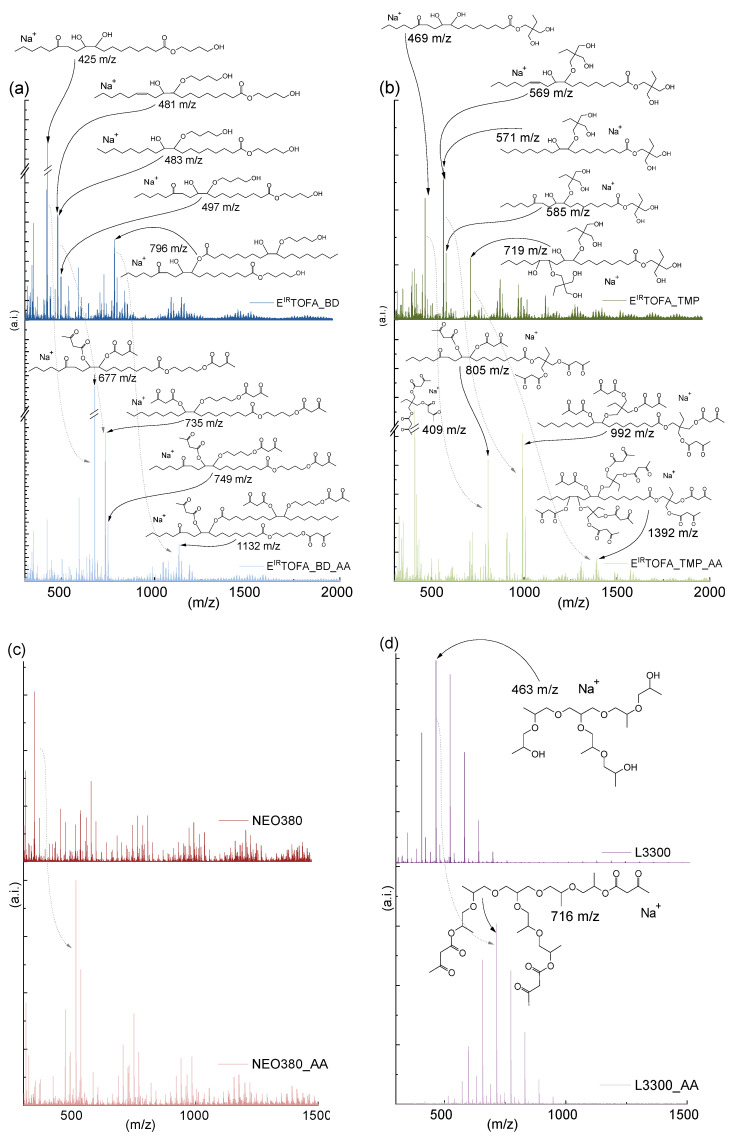
MALDI-TOF spectra of synthesized polyols and acetoacetylated polyols: (**a**) spectra of E^IR^TOFA_BD and E^IR^TOFA_BD_AA; (**b**) spectra of E^IR^TOFA_TMP and E^IR^TOFA_TMP_AA; (**c**) spectra of NEO380 and NEO380_AA; (**d**) spectra of L3300 and L3300_AA. For more information, see Appendix A Appendix A.

**Figure 5 polymers-14-04107-f005:**
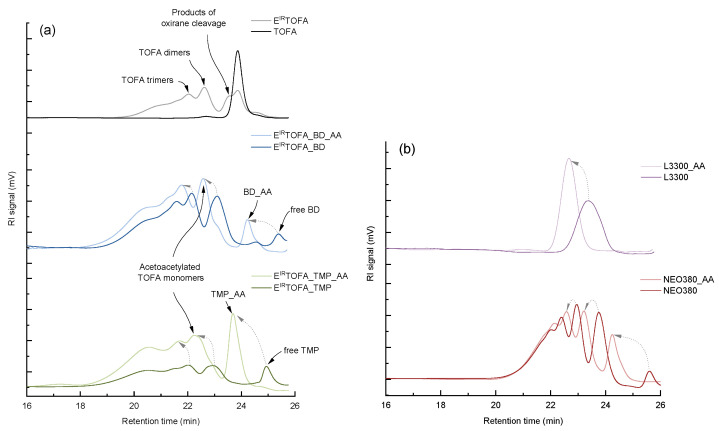
GPC/SEC chromatograms of TOFA, E^IR^TOFA, synthesized polyols, commercial polyols and their corresponding acetoacetate: (**a**) based on tall oils; (**b**) based on commercial polyols.

**Figure 6 polymers-14-04107-f006:**
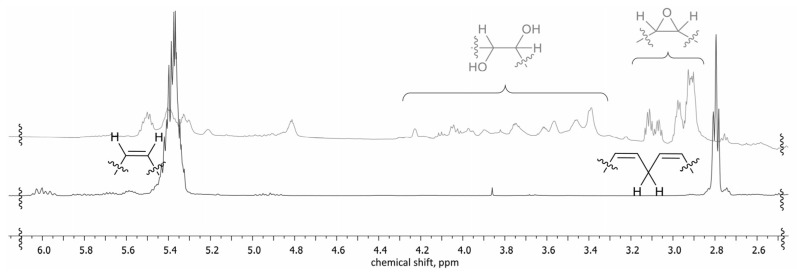
^1^H-NMR of TOFA and E^IR^TOFA (500 MHz, CDCl_3_).

**Figure 7 polymers-14-04107-f007:**
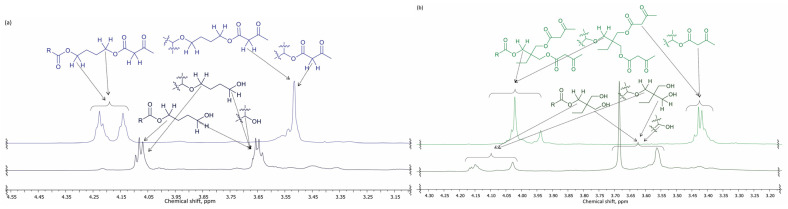
^1^H-NMR spectra of: (**a**) E^IR^TOFA_BD and E^IR^TOFA_BD_AA (500 MHz, CDCl_3_); (**b**) E^IR^TOFA_TMP and E^IR^TOFA_TMP_AA (500 MHz, CDCl_3_).

**Figure 8 polymers-14-04107-f008:**
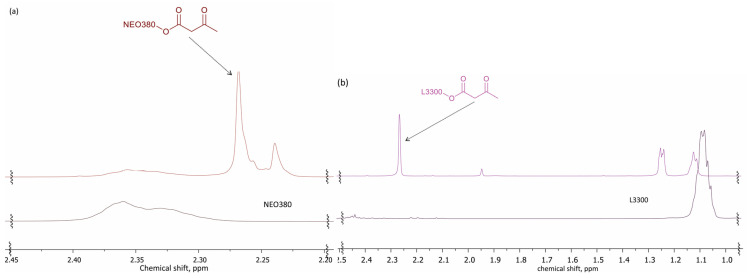
^1^H-NMR spectra of: (**a**) NEO380 polyol and NEO380_AA (500 MHz, CDCl_3_); (**b**) L3300 polyol and L3300_AA (500 MHz, CDCl_3_).

**Table 1 polymers-14-04107-t001:** The acid, hydroxyl, epoxy and iodine values, as well as moisture and acetoacetate group content of: synthesized tall-oil-based polyols and their acetoacetates, and commercial polyols and their acetoacetates.

Components		Acid Value, mg KOH/g	Hydroxyl Value, mg KOH/g	Iodine Value, g I_2_/100 g	Moisture, %	AA Groups, mol/100 g	Conversion of Hydroxyl, mol%
Tall-oil-based	TOFA	195 ± 3	-	157 ± 7	0.50 ± 0.02	-	-
	E^IR^TOFA	159 ± 2	-	52.4 ± 0.6	0.32 ± 0.01	-	-
	E^IR^TOFA_BD	5.8 ± 0.2	258 ± 5	-	0.20 ± 0.02	-	-
	E^IR^TOFA_BD_AA	<5	36.2 ± 0.8	-	0.025 ± 0.005	0.3307	80.5
	E^IR^TOFA_TMP	6.9 ± 0.4	415 ± 4	-	0.049 ± 0.007	-	-
	E^IR^TOFA_TMP_AA	<5	41.6 ± 0.3	-	0.037 ± 0.003	0.4562	83.7
Commercial polyols-based	NEO380	<5	371 ± 3	-	0.068 ± 0.002	-	-
	NEO380_AA	<5	40.7 ± 0.6	-	0.048 ± 0.002	0.4242	82.9
	L3300	<5	400 ± 5	-	0.060 ± 0.0012	-	-
	L3300_AA	<5	26.2 ± 0.4	-	0.021 ± 0.003	0.4456	89.5

**Table 2 polymers-14-04107-t002:** The rheological viscosity of synthesized components for tall-oil-based polyols and their acetoacetates, and for commercial polyols and their acetoacetates at a temperature of 25 °C.

Synthesized Components		Viscosity, mPa·s
Tall-oil-based	TOFA	27.0 ± 0.4
	E^IR^TOFA	3,012 ± 2
	E^IR^TOFA_BD	3,017 ± 5
	E^IR^TOFA_BD_AA	1,104 ± 3
	E^IR^TOFA_TMP	118,400 ± 20
	E^IR^TOFA_TMP_AA	5,466 ± 12
Commercial polyols-based	L3300	382 ± 2
	L3300_AA	211.1 ± 0.8
	NEO380	2,109 ± 8
	NEO380_AA	814 ± 3

## Data Availability

The raw data presented in this study are available on request from the corresponding author.

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
