# Peer review of "The Synthesis of Bio-Based Michael Donors from Tall Oil Fatty Acids for Polymer Development"

_polymers, 2022, doi:10.3390/polym14194107_

Round 1

Reviewer 1 Report

Title: Synthesis of Bio-based Michael Donors from Tall Oil Fatty Ac-2 ids for Polymer Development

Authors: Ralfs Pomilovskis, Inese Mierina, Hynek Beneš, Olga Trhlíková, Arnis Abolins, Anda Fridrihsone, Mikelis Kirpluks

Summary: in this work the authors present the synthesis of a Michael donor compound from tall oil fatty acids, a cellulose byproduct. The developed molecules were further used to obtain polymeric materials that might be an interesting alternative method to conventional isocyanate-based polyurethane materials.

Comments:

- FIgures 4, 5, 6, 7 and 8 should be improved, made bigger and provided with a better resolution and quality. In the current version it is impossible for the reader to actually check the spectra/GPC curves presented in the graphics.

- Authors state that: "MALDI-TOF spectra show that the acetoacetylation process has been successful", I agree with the observation but how was the reaction quantified? Which % of acetoacetylation was reached in the end of the reaction?

Reccomendation: I recommend publication of this manuscript after solving the minor comments listed above.

Reviewer 2 Report

The manuscript entitled “Synthesis of Bio-based Michael Donors from Tall Oil Fatty Acids for Polymer Development” refers to the synthesis Michael Donors with the use of natural compounds. Problem described in the manuscript is actual and interesting. In the article the new path of synthesis was proposed. The staring material is the tall oil. After epoxidation and oxirane ring opening with the use of two different alcohols – butane-1,4-diol and trimethylolpropane, polyols were esterificated with acetoacetates. Four products were characterized by chromatography, spectroscopies and analytical measurements.
I have some questions.
The scheme in Figure 1 is very communicative and explain the idea of the syntheses but synthesis procedures in the point 2.2 are not very clear, especially used abbreviations are ambiguous.
What about the statistical analysis of data in Table 1. At least standard deviations should be added.
The title of Table 1 is not completed. In the Table 1, there are given data refers to commercial components but the title does not included them.
Similarly situation in Table 2.
The viscosity of the new synthesized Michael donors is much more higher than the commercial polyol-based ones. Even, it is 6-times higher , especially in the case of EIRTOFA_TMP_AA. Is not it too high?
MALDI and SEC/GPC measurements pictured in Figures 4 and 5 show that obtained products are mixtures of many compounds. Is not the problem in the future applications?

The manuscript is basically well written and can be published after corrections.

Reviewer 3 Report

The manuscript by Pomilovskis et. al. reported the synthesis of Michael donor components from epoxidized tall oil fatty acids by ring-opening of oxirane and esterification reaction with different alcohols. This is a well-written manuscript. The authors have done careful work, and the presentation of the results is clear. However, there are several reports of epoxidation of tall oil fatty acid ring-opening and esterification reaction with different alcohols (e.g. Materials 2020, 13, 1985; doi:10.3390/ma13081985). The only new thing is that they synthesized Michael donor from tall oil fatty acids but failed to show further application of their synthesized Michael donor. The innovation of the current research work should be further highlighted and emphasized. The author should use those synthesized Michael donors for specific targeted reaction to validate their research. At the same time, the authors should consider the following comments to improve the quality of the paper. The authors are requested to make changes according to the comments below before the paper can be accepted for publication.

1.     The intensity of double bonds reduced during epoxidation but not completely vanished in the NMR. I will suggest the author to calculate the % conversion from the NMR

2. The quality of the figures must be improved.

Round 2

Reviewer 1 Report

Accept in present form.

Reviewer 2 Report

no additional comments

Reviewer 3 Report

The author addressed all the comments. The paper will be accepted in its present form.